# Language statistical learning responds to reinforcement learning principles rooted in the striatum

Joan Orpella[1], Ernest Mas-Herrero[2,3,4], Pablo Ripollés[1,5,6], Josep Marco-Pallarés[2,3,4], Ruth de Diego-Balaguer[2,3,4,7] *

1 Department of Psychology, New York University, New York, New York, United States of America, 2 Cognition and Brain Plasticity Unit, IDIBELL, L'Hospitalet de Llobregat, Barcelona, Spain, 3 Department of Cognition Development and Educational Psychology, University of Barcelona, Barcelona, Spain, 4 Institute of Neuroscience, University of Barcelona, Barcelona, Spain, 5 Music and Auditory Research Lab (MARL), New York University, New York, New York, United States of America, 6 Center for Language, Music and Emotion (CLaME), New York University, Max-Planck Institute, New York, New York, United States of America, 7 ICREA, Barcelona, Spain

* ruth.dediego@ub.edu

**Data Availability Statement:** All relevant data can be found in the paper's Supporting Information files and in https://neurovault.org/collections/10421/.

## Abstract

Statistical learning (SL) is the ability to extract regularities from the environment. In the domain of language, this ability is fundamental in the learning of words and structural rules. In lack of reliable online measures, statistical word and rule learning have been primarily investigated using offline (post-familiarization) tests, which gives limited insights into the dynamics of SL and its neural basis. Here, we capitalize on a novel task that tracks the *online* SL of simple syntactic structures combined with computational modeling to show that online SL responds to reinforcement learning principles rooted in striatal function. Specifically, we demonstrate—on 2 different cohorts—that a temporal difference model, which relies on prediction errors, accounts for participants' online learning behavior. We then show that the trial-by-trial development of predictions through learning strongly correlates with activity in both ventral and dorsal striatum. Our results thus provide a detailed mechanistic account of language-related SL and an explanation for the oft-cited implication of the striatum in SL tasks. This work, therefore, bridges the long-standing gap between language learning and reinforcement learning phenomena.

## Introduction

Statistical learning (SL) is the ability to extract regularities from distributional information in the environment. As a concept, SL was most popularized by the work of Saffran and colleagues, who first demonstrated infants' use of the transitional probabilities between syllables to learn both novel word forms [1] as well as simple grammatical relations (nonadjacent dependencies (NADs)) [2,3]. The idea of a mechanism for SL has since raised a considerable amount of interest, and much research has been devoted to mapping the scope of this cognitive feat. This work has been crucial in *describing* the SL phenomenon as it occurs across sensory

**Funding:** This work was supported by the European Research Council grant ERC-StG-313841 (TuningLang) (RdD-B), by the BFU2017-87109-P Grant from the Spanish Ministerio de Ciencia e Innovación (RdD-B), which is part of Agencia Estatal de Investigación (AEI) (Co-funded by the European Regional Development Fund. ERDF, a way to build Europe), and by ICREA Academia (JM-P). We also thank CERCA Program / Generalitat de Catalunya for the institutional support. The funders had no role in study design, data collection and analysis, decision to publish, or preparation of the manuscript.

**Competing interests:** The authors have declared that no competing interests exist.

**Abbreviations:** BOLD, blood oxygenation level–dependent; fMRI, functional magnetic resonance imaging; FWE, family-wise error; LLE, log-likelihood estimate; LLR, log-likelihood ratio; MNI, Montreal Neurological Institute; MTL, medial temporal lobe; NAD, nonadjacent dependency; PE, prediction error; RT, reaction time; RW, Rescorla-Wagner; SL, statistical learning; TD, temporal difference; VTA/SNc, ventral tegmental area/substantial nigra pars compacta.

modalities (auditory [4–6], visual [7,8] and haptic [9]), domains [8] (temporal and spatial), age groups [10,11], and even species (nonhuman primates [12] and rats [13]). After all this research, however, little is yet known about the *mechanisms* by which SL unfolds and their neural substrates.

One of the main reasons for this important gap in the literature is that the vast majority of SL studies have focused on the *output* of learning, generally assessed via offline post-familiarization tests, rather than on the learning process itself [14,15]. It is only recently that work on SL has started to shift toward the use of *online* measures of learning. Online measures afford a more detailed representation of the learning dynamics and thus offer the possibility of generating hypotheses about the computations underlying SL.

Online measures capitalize on the gradual development of participants' ability to *predict* upcoming sensory information (e.g., an upcoming syllable or word) as the regularities of the input are learned (e.g., a statistical word form or a grammatical pattern). Indeed, prediction is often understood as the primary *consequence* of SL [16,17]. Interestingly, however, the status of prediction as the driver of SL, rather than a mere consequence of it, i.e., its causal implication in learning, has not been explicitly investigated.

In the current study, we examined the online development of predictions as a fundamental computation for SL. In particular, we used an amply validated algorithm of reinforcement learning—temporal difference (TD) [18,19]—to model participants' online learning behavior and investigate its neural correlates. Note that, in adopting a model of reinforcement learning, a domain where reward generally plays an important role, we are not assuming (nor discarding) the phenomenological experience of reward (e.g., intrinsic reward [20,21]) during SL. Instead, we assessed whether particular computational principles reflected in TD learning can account for participants' SL behavior and their brain activity during learning.

TD models are based on the succession of predictions and prediction errors (the difference between *predicted* and *actual* outcomes) at each time step, by which predictions are gradually tuned. In contrast to models typically used to explain SL (e.g., [22,23]), a vast body of research supports the neurobiological plausibility of TD learning, with findings of neural correlates of predictions and prediction errors both using cellular-level recordings and functional magnetic resonance imaging (fMRI). Several brain areas, notably the *striatum*, have been implicated in the shaping of predictions over time and the selection of corresponding output behavior [24–29]. Interestingly, activity in the striatum has also been documented in the SL of NADs [30,31] as well as of phonological word forms [32], but the precise role of these subcortical structures in this domain remains unspecified.

With the aim of clarifying the mechanisms for SL and their neural underpinnings, we combined computational (TD) modeling with fMRI of participants' brain activity while performing a language learning task. In particular, participants completed an incidental NAD learning paradigm. In natural languages, NADs are abundant and underly important morphological and syntactic probabilistic rules (e.g., the relationship between *un* and *able* in *un*believ*able*). Sensitivity to NADs is therefore important in the early stages of grammar learning, when the relation between phrase elements is tracked at a superficial level and before more abstract representations (syntactic rules) can be created via other mechanisms and brain structures [33]. However, sensitivity to NADs can also be critical for speech segmentation in the early stages of word learning [34], both in prelexical development [2] and beyond (i.e., second language acquisition [35]).

The main advantage of this particular SL task over similar tasks (e.g., [2,36]) is that it provides a reliable measure of *online* learning [37] that we can then model. For modeling, we used a TD algorithm for its greater sensitivity to temporal structure compared to simpler RL models (e.g., Rescorla-Wagner (RW) [38]). Note that this an important prerequisite for NAD learning,

since the to-be-associated elements are separated in time. Nonetheless, we additionally compared the adequacy of these simpler algorithms to that of the TD model.

We expected the interplay of predictions and prediction errors, as modeled by the TD algorithm, to closely match participants' online SL behavior. In addition, and in line with the aforementioned research on both reinforcement learning and SL, we expected striatal activity to be associated with the computation of predictions.

## Results

Two independent cohorts (behavioral group: $N = 19$; fMRI group: $N = 31$) performed the same incidental NAD learning task (Fig 1; see Materials and methods for details). In brief, participants were auditorily exposed to an artificial language, which, unbeknown to them, contained statistical dependencies between the initial (A) and final (C) elements of 3-word phrases with variable identity of middle X elements. Orthogonal to SL, participants' instructions were to detect the presence or absence of a given target word, which was always the final C element of one of the 2 A_C dependencies presented (Fig 1). The online SL of the NADs was measured as participants' decrease in reaction times (RTs) over trials, which reflects the gradual learning of the predictive value of the initial element A in respect to the dependent element C of each phrase (i.e., the equivalent of learning that *un* predicts *able* in *un*believ*able*). In line with previous research [33,37], we expected faster RTs in a so-called NADs block with such dependencies compared to a Random block with no statistical dependencies (i.e., equally probable element combinations). This indicates that participants learned the dependency between A and C elements and were thus able to use the identity of the initial word A to predict the presence or absence of their target word C.

This behavioral paradigm was initially tested in a group of 19 volunteers (behavioral group: $N = 19$; 15 women; mean age = 21 years, $SD = 1.47$). After ruling out Order effects (NADs block first/Random block first; main effect of Order and all its interactions with other factors $p > 0.4$), a repeated measures ANOVA with Structure (NADs/Random) and Target (Target/No Target) as within-participant factors confirmed that SL of the dependencies occurred over the NADs block. In particular, responses to phrases in the NADs block were overall faster compared to the Random block ($F(1,18) = 13.6$, $p < 0.002$, Partial $\eta^2 = 0.43$; mean difference = 149.40 ms, $SE = 40.51$). A significant effect of Target ($F(1,18) = 24.46$, $p < 0.001$, Partial $\eta^2 = 0.58$) further indicated, as reported previously [37], that responses to target C elements were faster than to no target C elements (mean difference = 68.66 ms, $SE = 13.88$). Importantly, we found no interaction between Structure and Target ($F(1,18) = 0.53$, $p > 0.48$), suggesting that both dependencies were learned comparably.

These results were furthered using a linear mixed model analysis. This analysis allows us to assess whether learning occurred progressively over trials, by using the RTs for all trials and participants. Note that this represents a more sensitive measure of the *online* SL process than a single mean value per participant per block. Specifically, we contrasted the slopes for each of the main conditions (NADs versus Random). A steeper slope (i.e., more negative; Fig 2A) for the NADs block compared to the Random block (ßdiff = −5.84, t = −8.88, $p < 0.001$; ßdiff is the estimate of the difference in slopes between the NADs and Random conditions; see Materials and methods) indicated that learning of the dependencies occurred gradually over the trials of the NADs block.

We next assessed the extent to which a TD model (see Materials and methods) could predict participants' SL behavior. If participants' RTs reflect SL in terms of the ability to *predict* the last element (C) of a phrase from the identity of the initial (A) element, their overall development should be mimicked by the model parameter representing the predictive value of the

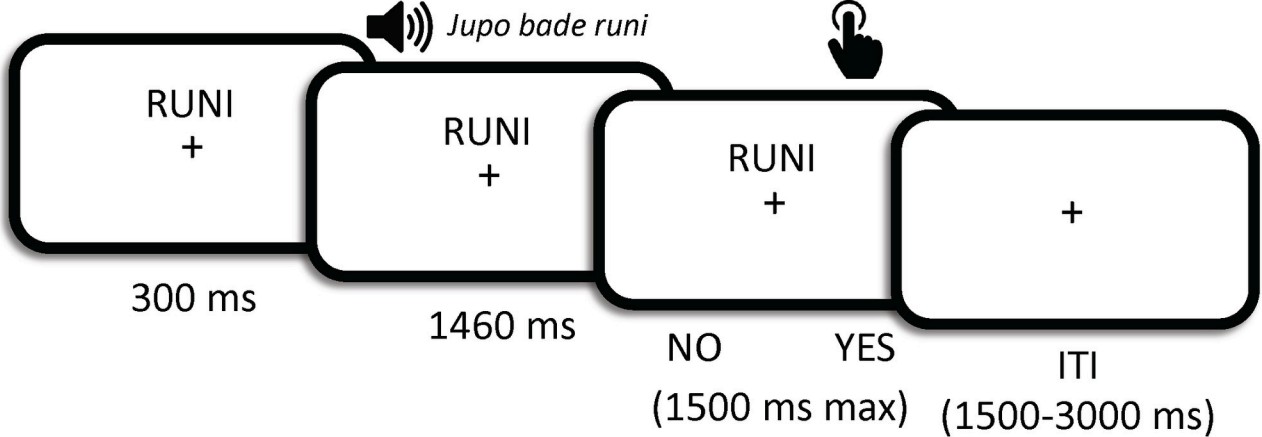

**Fig 1. NAD learning task.** The same procedure was used for the behavioral and fMRI groups, except for the ITI, lasting 1 second in the behavioral group and jittered between 1.5 and 3 seconds in the fMRI group. Participants were requested to indicate, via a button press, whether the target word was present (YES) or absent (NO) after listening to each phrase, with a maximum of 1.5 second allowed for response. RTs were calculated from the onset of the third word of each phrase. Trial (i.e., event) onsets for fMRI analyses were set to the beginning of each phrase. fMRI, functional magnetic resonance imaging; ITI, intertrial interval; NAD, nonadjacent dependency; RT, reaction time.

initial (A) element ($P(A)$; see Materials and methods). That is, with time, the predictive value of the initial A element according to the TD model should change (*increase*) in a way resembling participants' RTs. Note that RTs can reflect prediction learning as well as fluctuations due to decision processes, motor response preparation and execution, random waxing and waning of attention, and system noise, which are not the object of this investigation. Indeed,

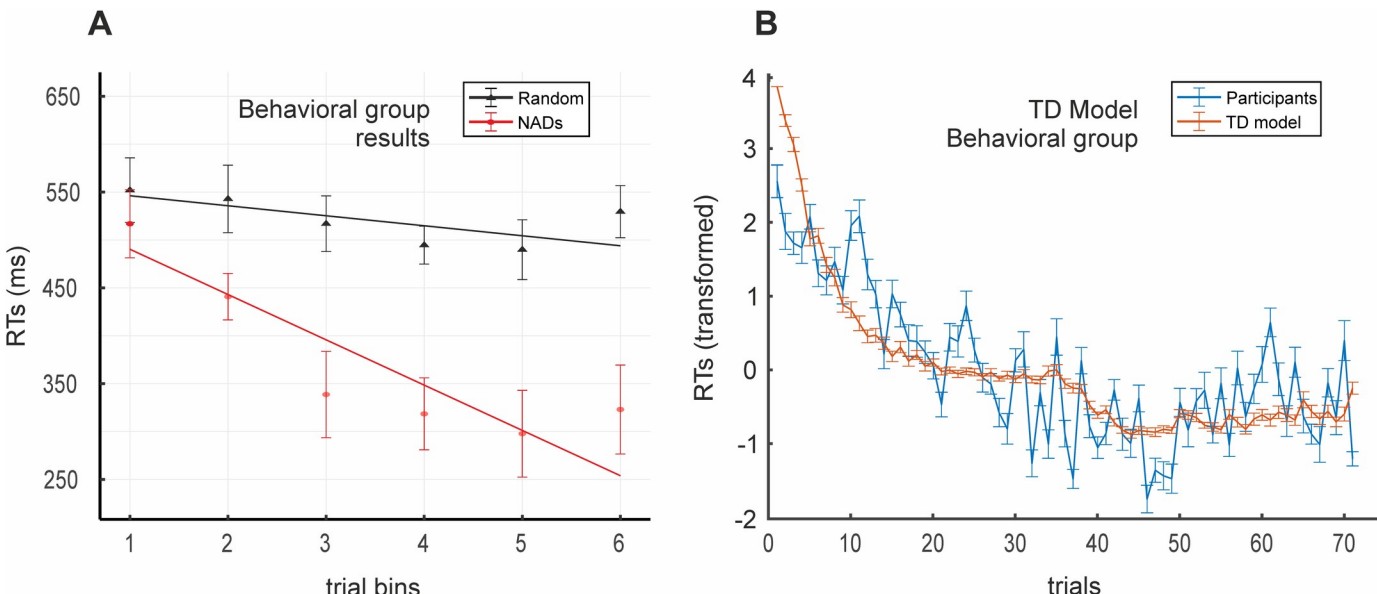

**Fig 2. Behavioral group's SL and TD model results. (A)** Slopes for NADs and Random blocks ($N = 19$) derived from the linear mixed model. A significantly steeper slope for the NADs block (red) compared to the Random block (black) indicates SL over trials. Actual data shown averaged into 6 trial bins with the SEM for display purposes only. All data points per participant were used for the linear mixed model analysis. **(B)** Plot of participants' mean RTs (blue) against the TD model's estimates of the development of predictions over learning (red; inverted as 1-$P(A)$ before averaging and z-scoring for display purposes). Vertical bars = SD. RTs were initially transformed (see Materials and methods) and are plotted with the inverted model estimates in z-score values. $P(A)$ = TD model's predictions from the initial word (A) of the dependencies. Data used to generate Fig 2 can be found in S1 Data. NAD, nonadjacent dependency; RT, reaction time; SL, statistical learning; TD, temporal difference.

we used modeling to strip these off and so derive a purer measure of prediction learning. Fig 2B shows the development of participants' RTs over trials plotted against the development of the predictive value of the initial word A (*P*(A)) as computed by the TD model (inverted as 1-*P* (A) and z-scored for display purposes). Model fit was evaluated at the individual level by the model fit index (see Materials and methods), calculated as 1 minus the log-likelihood ratio (LLR) between the log-likelihood estimate (LLE) for the TD model and the LLE for a model predicting at chance. Model fit index values of 1 would indicate an exact model fit. Our results show a group average model fit index of 0.74 (*std* = 0.05), indicating that the TD model was 3 to 4 times better than the chance model at adjusting to participants' RTs. We additionally compared the performance of the TD model against that of an RW model [38] (S1 Fig). In contrast to the TD model, the RW model treats each AX_ combination as a single event, therefore combining the predictive values of the 2 (A plus X) elements [38–40], and so does not take into account nonadjacent relations (which are also captured by the TD model through the devaluation of the prediction parameter; see Materials and methods). A paired-samples *t* test indicated that model fit index values produced by the TD model were significantly better than those produced by the RW model (mean difference = 0.098, *SE* = 0.013; $t(18) = 7.65$, $p < 0.001$, *d* = 0.83), which only achieved an average model fit index of 0.64 (2 to 3 times better than the chance model; *std* = 0.1). The TD model was, therefore, superior to the RW in adjusting to each participant's RT data.

We then replicated these behavioral results on a new cohort of participants from whom we additionally acquired fMRI data while performing the incidental NAD learning task (fMRI group; *N* = 31; 20 women; mean age = 23 years, *SD* = 3.62). We used the same analytical procedure to evaluate SL at the behavioral level and model adequacy thereafter. Having discarded block order effects (main effect of Order and all interactions: $p > 0.1$), a repeated measures ANOVA with factors Structure (NADs/Random) and Target (Target/No Target) indicated that SL occurred in the NADs block ($F(1,30) = 4.96$, $p < 0.034$, Partial $\eta^2 = 0.14$), again with faster mean RTs to phrases in the NADs block compared to the Random block (mean difference = 42.67 ms, *SE* = 19.16). As expected, RTs to target C elements were faster than to no target C elements (mean difference = 57.2 ms, *SE* = 43.28; $F(1,30) = 54.15$, $p < 0.001$, Partial $\eta^2 = 0.64$). As with the behavioral group's data, the null interaction between the factors Structure and Target ($F(1,30) = 0.168$, $p > 0.68$) indicated that both target and no target dependencies were similarly learned. A linear mixed model again indicated gradual learning of the dependencies, with a steeper slope for the NADs block compared to the Random block (ßdiff = −2.4, $t = −6.9$, $p < 0.001$; Fig 3A).

We next fitted the TD model to the fMRI group behavioral dataset. The development of participants' RTs is plotted in Fig 3B with the development of the predictive value of the initial element (A) according to the TD model. At a group level, the mean model fit index was 0.71 (*std* = 0.04), again indicating a fit between 3 and 4 times better than that of a model predicting at chance. This was also significantly better than the average model fit index produced by the RW model (mean difference = 0.11, *SE* = 0.01; $t(30) = 12.20$, $p < 0.001$, *d* = 1.26), which only reached a benchmark of 0.6 (again, 2 to 3 times better than the chance model; *std* = 0.08; S1 Fig).

These results, therefore, represent a replication of our previous findings from the behavioral group, both in terms of participants' overall SL behavior and of the adequacy of the TD model in providing a mechanistic account of its dynamics.

To investigate the brain areas or networks sensitive to the trial-wise computations related to SL from speech, we used a measure of the trial-by-trial development of predictions from the initial word A of structured phrases (*P*(A); see Materials and methods) as estimated by the TD model for each participant. Specifically, we correlated this proxy for prediction learning with

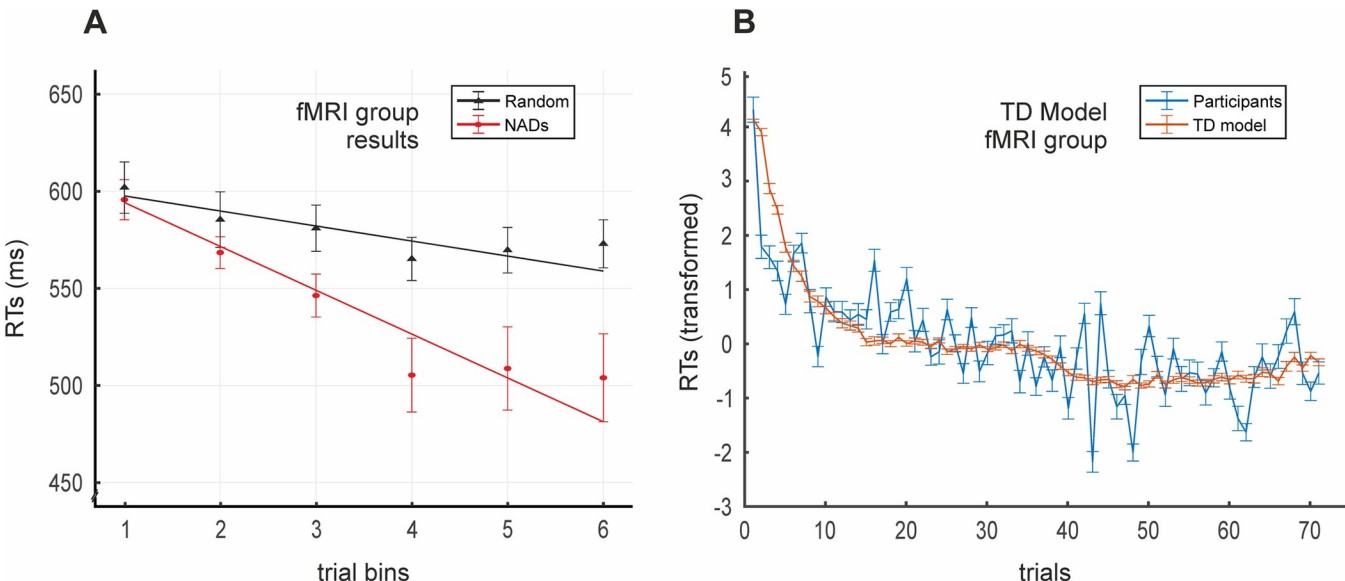

**Fig 3. fMRI group's SL and TD model results.** (A) Slopes for NADs and Random blocks (*N* = 31) derived from the linear mixed model. A significantly steeper slope for the NADs block (red) compared to the Random block (black) indicates SL over trials. Actual data shown averaged into 6 trial bins with the SEM for display purposes. All data points per participant were used for the linear mixed model analysis. (B) Plot of participants' mean RTs (blue) against the TD model's estimates of the development of predictions over learning (red; inverted as 1-*P*(A) before averaging and z-scoring for display purposes). Vertical bars = SD. RTs were initially transformed (Materials and methods) and are plotted with the inverted model estimates in z-score values. *P*(A) = TD model's predictions from the initial word (A) of the dependencies. Data used to generate Fig 3 can be found in S2 Data. fMRI, functional magnetic resonance imaging; NAD, nonadjacent dependency; RT, reaction time; SL, statistical learning; TD, temporal difference.

each participant's trial-wise blood oxygenation level–dependent (BOLD) signal measures for the NADs block, time locked to the onset of the A element of each phrase. We only report results for clusters family-wise error (FWE)-corrected at *p* < 0.001 at the cluster level (*p* < 0.001 uncorrected at the voxel level; minimum cluster size = 20). The contrast between *P*(A)-modulated NADs block activity against an implicit baseline (see Materials and methods) yielded a large cluster covering most of the striatum (i.e., bilateral caudate nuclei, putamen, and ventral striatum; Fig 4 and S1 Table). Also, noteworthy, there were 2 additional clusters, one in the left superior posterior temporal gyrus extending medially to Rolandic opercular regions and another including right inferior and middle occipital areas. While formalized as prediction learning, we note that activity in these regions could also reflect the gradual increase in prediction error on the initial element A of each phrase. This is because the specific A element can never be anticipated, and, therefore, predictions and prediction errors should be commensurate with each other. An investigation of prediction error responses on the C (target/no target) elements was not possible due to the presence of button presses on these elements as required by the task.

In order to further support the specificity of these results, we completed a series of control analyses. First, while, by definition, no structure can be derived from the Random block, this does not preclude the engagement of particular brain regions in the attempt to capture the relationship between specific phrase elements. In other words, we cannot ascertain that similar type computations are not taking place in the Random block, even when these will accrue no substantial knowledge. Therefore, to assess the specificity of the reported clusters in prediction *learning*, we next contrasted the *P*(A)-modulated activity for the NADs block with the equivalent in the Random block (*P*(X1)) directly (see Materials and methods). That is, we compared the brain activity related to the trial-by-trial predictive value of stimuli A during the NADs

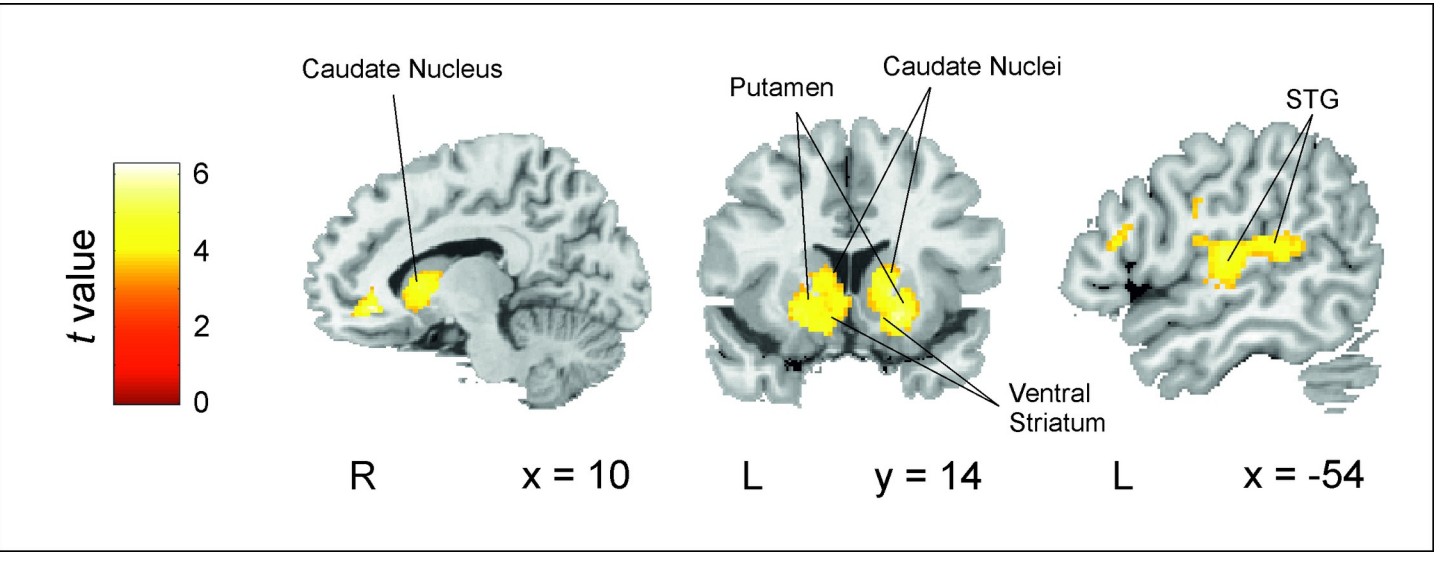

**Fig 4. Brain regions related to changes in the predictive value of the initial word of each phrase in the NADs block vs. implicit baseline (i.e., NADs *P*(A)-modulated activity vs. implicit baseline).** Activity in the basal ganglia (bilateral caudate nuclei, putamen, and ventral striatum) and in the left posterior STG was modulated by the trial-by-trial development of predictions (*P*(A)) as estimated by the TD model (contrast: NADs *P*(A)—Baseline). Results are reported for clusters FWE-corrected at $p < 0.001$ at the cluster level (minimum cluster size = 20). Neurological convention is used with MNI coordinates shown at the bottom right of each slice. Data used to generate Fig 4 can be found in http://identifiers.org/neurovault.collection:10421. FWE, family-wise error; NAD, nonadjacent dependency; STG, superior temporal gyrus; TD, temporal difference.

block to its counterpart during the Random block (i.e., X1). Significant differences centered on the same 3 relatively large clusters (S2 Fig and S2 Table) observed in the main analysis, namely bilateral caudate, putamen and ventral striatum, left transverse and posterior superior temporal gyri, and right middle occipital cortex (not shown in the figure). The converse contrast (*P*(X1)-modulated Random versus *P*(A)-modulated NADs) did not produce any significant results.

It is generally understood that the final goal of (TD) learning is to inform behavior [40]. Even if we consider predictions themselves as some form of *covert* behavior [41] used to optimize online learning and processing, our paradigm also required participants to make an *overt* response (a button press) to the presence of their target word. RTs are often understood as modulators of a condition's related BOLD signal measurements and used to extract the variability pertaining to such motor responses. However, as previously illustrated (Figs 2 and 3), RTs in the NADs block will tend to show a close relationship to online learning. Hence, a more suitable baseline to remove response-related brain activity is the RTs to the Random block, i.e., where no SL can occur. We therefore contrasted *P*(A)-modulated NADs block activity with the RT-modulated Random block activity (activity estimates for the contrast between *P*(A)-modulated NADs and RT-modulated NADs activity are also reported in S4 Fig and S4 Table). Significant prediction-related NADs activity remained in the dorsal striatum, particularly in bilateral caudate nuclei and right putamen (S3 Fig and S3 Table).

Finally, we investigated the brain areas related to SL at the block level rather than to online (trial-by-trial) SL. Specifically, for each participant, we computed the difference between their mean RT for the Random block and their mean RT for the NADs block (we called this the NADs effect). Larger NADs effect values indicate more aggregate SL. We then covaried, at the second level of analysis, each participant's NADs block minus Random block first-level contrast with their own NADs effect. As can be seen in Fig 5 (S5 Table), activity centered on 2 large clusters. One cluster covered much of the left frontal insular cortex reaching the

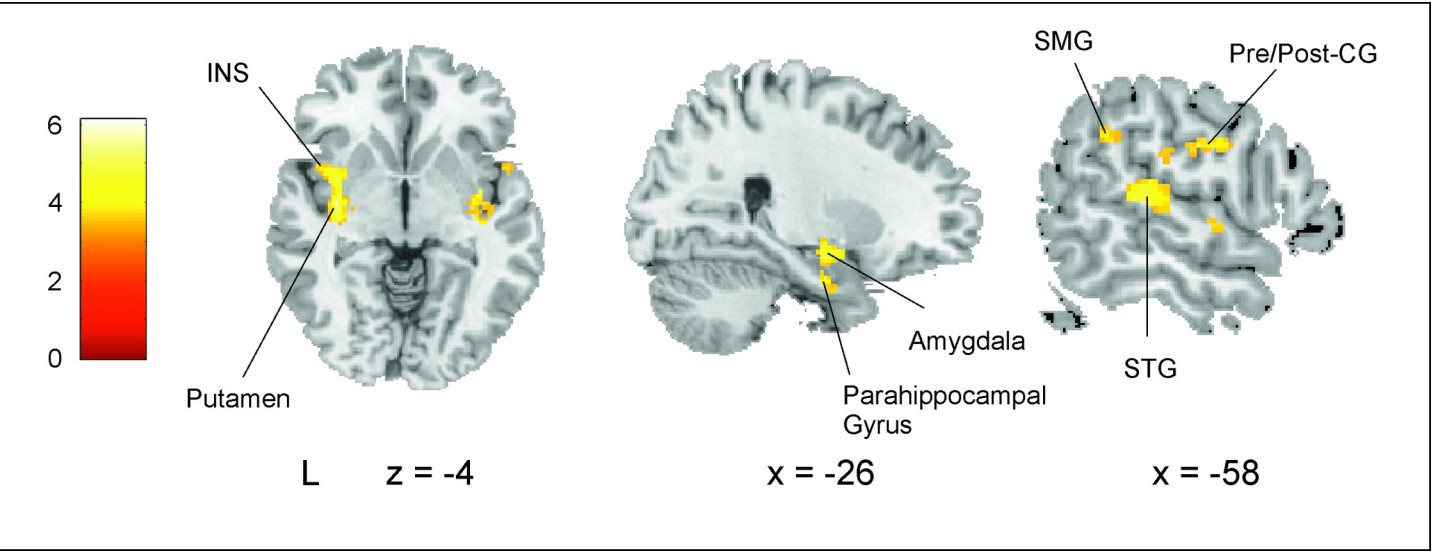

**Fig 5. Brain regions related to mean SL (i.e., NADs block vs. Random block activity covarying with the NADs effect).** Significant brain activity for the NADs block minus the Random block covarying with the NADs effect, i.e., an aggregate measure of SL. Results are reported for clusters FWE-corrected at $p < 0.001$ at the cluster level (minimum cluster size = 20). MNI coordinates were used. Data used to generate Fig 5 can be found in http://identifiers.org/neurovault.collection:10421. BA, Brodmann area; CG, central gyrus; FWE, family-wise error; INS, insula; NAD, nonadjacent dependency; SL, statistical learning; SMG, supramarginal gyrus; STG, superior temporal gyrus.

putamen, amygdala, and anterior parahippocampal gyrus. The second extended from the left superior temporal gyrus through the Rolandic operculum to the pre/post-central gyri and insula.

Altogether, therefore, our analyses (main and control) demonstrate that activity within the striatum was related to the computations that specifically facilitate online statistical NAD learning from speech as predicted by the TD model.

## Discussion

In this study, we provide evidence for the SL of NADs as an instance of reinforcement learning. A TD model of reinforcement learning, which capitalizes on the iteration of predictions and prediction errors, was able to mimic participants' RT data reflecting gradual SL over trials. This was replicated on 2 independent cohorts, producing similar model fits that were also clearly superior to those of simpler learning models. Functional neuroimaging data of participants' online learning behavior also allowed us to examine the neural correlates of prediction-based SL. In line with neurocomputational models of TD learning, the trial-by–trial development of predictions from the initial word of the dependencies was strongly related to activity in bilateral striatum. Importantly, striatal activity was unrelated to the overt motor responses required by the task (i.e., button presses) or more general computations, supporting the implication of the striatum specifically in prediction-based SL.

Evidence for the adequacy of a TD algorithm in capturing participants' online NAD learning behavior offers novel insights into the mechanisms for SL. In particular, our results underscore the causal role of predictions for learning, compelling us to reassess the commonly assumed relationship between SL and predictive processing. Indeed, SL not only enables predictions (predictions as *a consequence* of SL), as generally understood (see p.e. [14]), but also capitalizes on predictions (predictions as *a cause* of SL). This new understanding of SL can

thus offer interesting reinterpretations of previously reported correlations between SL abilities and predictive processing [16], raising questions about the direction of causality.

Moreover, our results make an important contribution to the understanding of the neurobiological basis of SL. While previous research [14,42] has shown a similar behavioral development of online SL (cf. Fig 2), brain imaging data and its link to a mechanistic explanation of learning were lacking. Here, we used a measure of online SL behavior in combination with computational modeling and fMRI data to unveil the basic mechanism underlying learning and its brain correlates. A complementary approach to describing online SL, which involves the frequency tagging of participants' neurophysiological responses over learning [43–45], has recently been used to track the emergence of new representations (in time and neuroanatomical space) as participants learn. We add to these findings by providing a mechanistic account for *how* these representations (i.e., learning) come to be and a plausible neuroanatomical substrate for its key computations. In particular, we show that the gradual development of predictions for SL is related to robust and widespread activity in bilateral striatum (Fig 4). This finding adds a valuable degree of specificity to the oft-cited implication of these subcortical structures in artificial grammar learning and SL more generally [30–32].

Both the adequacy of a TD model and the involvement of the striatum in prediction-based SL place this cognitive ability squarely in the terrain of reinforcement learning. Indeed, the link between prediction learning and activity in the striatum is one of the most robust findings in the reinforcement learning literature, from intracranial recordings to fMRI studies [25–29,40,46–48]. Activity in the ventral striatum, in particular, has been associated with the delivery and anticipation of rewarding stimuli of different types (i.e., from primary to higher-order rewards) [49]. More specifically, the ventral striatum interacts in complex ways with the dopaminergic system (mainly ventral tegmental area/substantial nigra pars compacta (VTA/SNc)) with responses consistent with the computation of reward prediction error [24,50–52]. Under this light, our reported pattern of activity in the ventral striatum is consistent with the gradual transfer over learning of prediction error–related dopaminergic responses from rewarding to predictive stimuli as found in classic conditioning paradigms [53,54]. That is, a gradual increase in response on A elements may be expected as their predictive value is learned, since these elements can never be anticipated. Alternatively, activity in the ventral striatum could reflect inhibitory signals aimed to attenuate dopaminergic inputs from the VTA/SNc [55] in response to C elements as these become more predictable.

From a theoretical standpoint, it may be necessary to distinguish between the response of the reward system for learning and the phenomenological experience of reward [21,56]. Recent evidence [57–59], nonetheless, supports the notion of language learning as *intrinsically* rewarding [20] and suggests quantitative over qualitative differences between endogenous and exogenous sources of reward [21]. So far, the adequacy of reinforcement learning algorithms for the learning of intrinsically rewarding tasks has mainly remained theoretical [21,60]. Our results now contribute to this literature by showing their suitability in specific instances of SL.

Still within the computations of the TD model, activity in the dorsal striatum (caudate and putamen) responds to the updates at each time step of the outcome value representations associated with each stimulus in ventromedial and orbitofrontal areas [61]. By updating the value associated with a particular behavioral option, the dorsal striatum takes a leading role in the selection of the most appropriate behavior [62,63]. In agreement with our results, recent data [64] suggest that this role also applies to the language domain, with evidence for an implication of the caudate nucleus in the selection of linguistic alternatives from left prefrontal perisylvian regions with which it is connected [65]. The caudate could also promote the *attentional* selection of behaviorally relevant elements from frontal cortical areas [66–68]. In our case, while this attentional selection should initially pertain to C elements (the target of the monitoring

task), a shift toward A elements may also be expected as their predictive value increases. This interpretation is consistent with the finding of a gradual increase in an early attentional event-related component (the P2) over the exposure to (the A elements of) NADs but not to similar but unstructured material [69,70].

The appropriate behavioral alternative, in our case, pertaining to the identity of the final C element of each phrase, may be finally concretized as a specific motor articulatory representation of the element selected by the putamen from speech pre/motor areas used to predict upcoming auditory input [41]. This selection of a "covert" motor response is consistent with the attenuated (though not eliminated) activity in the putamen when regressing out overt motor responses (button presses; S3 Fig; cf. Fig 4). This cascade of processes, with the final selection of motor articulatory representations by the putamen, may be used to generate the corresponding auditory predictions [41], ultimately translating into increasingly faster RTs for the predicted C elements. In this view, activity in the posterior superior temporal gyrus (Fig 4) would reflect the downstream (i.e., sensory) consequences of this selection [41,71]. It is unknown at this moment in which representational space (e.g., auditory, motor, somatosensory), or by which mechanism, actual prediction *testing* takes place. However, the present data suggest that prediction-based SL may be fundamentally linked to such motor engagement as part of a learning mechanism orchestrated by the striatum. This is consistent with the observation that participants that are better at predicting speech inputs embedded in noise, a situation known to involve the speech motor system [72], are also better statistical learners [16], and agrees with the well-accepted role for these structures in procedural learning [73,74] and the managing of motor routines [52,65,75]. We speculate that this prediction mechanism via motor articulatory representations should become of critical importance for learning when putative alternative learning mechanisms (e.g., hippocampus; see below) are weakest, for example, when a temporal separation is imposed between the elements to be associated, as in our NAD learning task.

It is interesting that, in contrast to recent findings [76], we did not find an implication of the hippocampus/medial temporal lobe (MTL) related to the *online* SL of NADs. Hippocampal/MTL and basal ganglia activity are often thought to reflect the workings of 2 distinct (complementary or competing) learning systems, traditionally related to declarative or explicit versus procedural or implicit learning systems, respectively [77]. Although striatal activity in our study is consistent with the incidental nature of our SL task, other SL tasks of incidental learning have also reported MTL/hippocampus [78] engagement. As mentioned previously, the difference may owe to the type of statistical relations present in the material. Specifically, as reflected in biologically inspired computational models (e.g., [78,79]) and in line with recent data [45], the MTL/hippocampus appears to capitalize on the relationship between pairs of adjacent stimuli. This contrasts with the TD model, which learns in part due to the low TPs between adjacent (AX and XC) elements (cf. [80]), and deals explicitly with the NADs through a temporal discounting parameter. It is thus unclear how the aforementioned models of the hippocampus would fare in the SL of NADs, where the relationship between adjacent elements is very weak. As it occurs with declarative and procedural learning, nevertheless, it is likely that these different learning systems be concurrently engaged in new learning situations. Reports of hippocampal or striatal activity in SL tasks will therefore depend on the nature of the materials employed. Finally, hippocampal activity may also respond to memories of the units of SL [81,82] (e.g., words in our case) as well as to its outputs [83]. Under this light, activity in the MTL system should correlate with aggregate measures of SL, as we observe (Fig 5), rather than with *online* SL. This is also in line with recent data [45] relating hippocampal activity to the encoding of the output units of SL (e.g., 3-syllable words), contrary to activity in prefrontal regions, as part of the frontostriatal system, which tracked the TPs. The striatum, in contrast,

appears to be in charge of probabilistic learning [84] and may therefore be required in situations where uncertainty is present [85], as in our task. From this standpoint, the outputs of each system could thus potentially feed into each other; i.e., while the striatal system might utilize representations stored in the hippocampal system for SL, the latter might also come to store the outputs of that SL.

In sum, by the combination of an online measure of SL, computational modeling, and functional neuroimaging, we provide evidence for SL as a process of gradual prediction learning strongly related to striatal function. This work, therefore, makes a valuable contribution to our understanding of the mechanisms and neurobiology of this cognitive phenomenon and introduces the provoking possibility of language-related SL as an instance of reinforcement learning orchestrated by the striatum.

## Materials and methods

### Participants

Two independent cohorts participated in the study. We first collected data from 20 volunteers from the Facultat de Psicologia of the Universitat de Barcelona as the behavioral group. Data from 1 participant were not correctly recorded, so the final cohort comprised 19 participants (15 women, mean age = 21 years, $SD$ = 1.47). We used the partial $\eta^2$ obtained for the main effect in the NADs block in the behavioral group to compute a sample size analysis for the fMRI group. To ensure 90% of power to detect a significant effect in a $2 \times 2$ repeated measures ANOVA at the 5% significance level based on this measure of effect size, MorePower [86] estimated that we would need a sample size of at least 16 participants. However, considering (i) that we expected participants to perform worse inside of the fMRI scanner and (ii) the recommendation that at least 30 participants should be included in an experiment in which the expected effect size is medium to large [87], we finally decided to double the recommended sample size for the fMRI experiment. The fMRI group thus consisted of 31 participants (20 women, mean age = 23 years, $SD$ = 3.62) recruited at the Universidad de Granada. All participants were right-handed native Spanish speakers and self-reported no history of neurological or auditory problems. Participants in the fMRI group were cleared for MRI compatibility. The ERC-StG-313841 (TuningLang) protocol was reviewed and monitored by the European Research Council ethics monitoring office, approved by the ethics committee of the Universitat de Barcelona (IRB 00003099), and conducted in accordance with the Declaration of Helsinki. Participation was remunerated and proceeded with the written informed consent of all participants.

### Statistical learning paradigm

Two different artificial languages were used in the NAD learning task. Each language comprised 28 bisyllabic (consonant-vowel-consonant-vowel) pseudowords (henceforth, words). Words were created using Mbrola speech synthesizer v3.02b through concatenating diphones from the Spanish male database "es1" (https://github.com/numediart/MBROLA) at a voice frequency of 16 KHz. The duration of each word was 385 ms. Words were combined to form 3-word phrases with 100 ms of silence inserted between words. Phrase stimuli were presented using the software Presentation (Neurobehavioral Systems, Berkeley, CA, USA) via Sennheiser over-ear headphones (pilot group) and MRI-compatible earphones (Sensimetrics, Malden, MA, USA; fMRI group).

The learning phase consisted of a NADs block and a Random block, each employing a different language. The order of blocks was counterbalanced between participants. We also counterbalanced the languages assigned to NADs and Random blocks. The NADs block consisted of 72 structured phrases (phrases with dependencies) whereby the initial word (A) was 100%

predictive of the last word (C) of the phrase. We used 2 different dependencies (A1_C1 and A2_C2) presented over 18 different intervening (X) elements to form AXC-type phrases. Twelve of the 18 X elements were common to both dependencies, while the remaining 6 were unique to each dependency. These 36 structured phrases were presented twice over the NADs block, making a total of 72 AXC-type structured phrases issued in pseudorandom order. The probability of transitioning from a given A element to a particular X was therefore 0.056. Phrases in the Random block were made out of the combination 2 X elements and a final C element (either C1 or C2, occurring with equal probability). Note that, while C elements could be predicted with 100% certainty in the NADs block, these could not be predicted from the previous X elements in the Random block. X elements were combined so that each X word had an equal probability to appear in first and second position but never twice within the same phrase. Forty-eight unstructured phrases were presented twice over the Random block, making a total of 96 pseudorandomized XXC-type unstructured phrases. Each 3-word phrase, in both NADs and Random blocks, was considered a trial for the fMRI analysis. A recognition test was issued at the end of each block to assess offline learning (see S1 Text for further details).

To obtain an online measure of incidental learning, participants were instructed to detect, as fast as possible via a button press, the presence or absence of a given target word. The target word for each participant remained constant throughout the block and was no other than one of the 2 C elements of the language (C1 or C2, counterbalanced). A written version of the participant's target word was displayed in the middle of the screen for reference throughout the entire learning phase. Importantly, participants were not informed about the presence of dependencies, so this word-monitoring task was in essence orthogonal to SL. Yet, if incidental learning of the dependencies occurred over trials in the NADs block, faster mean RTs should be observed for this block compared to the Random block where the appearance or nonappearance of the target word could not be anticipated from any of the preceding elements.

Fig 1 details a trial in the SL task. The participant's target word (e.g., RUNI) appeared on the screen above a fixation cross to signal the beginning of each trial and remained on the screen throughout the trial. A 3-word phrase (1,460 ms) was delivered auditorily 300 ms later, followed by a prompt to respond YES/NO to target presence/absence, respectively. A maximum of 1,500 ms was given for participants to indicate their response before the intertrial interval began. Upon response, the target word disappeared into the intertrial interval, which lasted 1,000 ms (behavioral group) or was jittered between 1,500 ms and 3,000 ms (fMRI group). fMRI analyses (event onsets; see below) were time locked to the onset of each phrase presentation, and the trial duration was defined as the duration of the phrase.

Participants in the behavioral group indicated the detection or nondetection of the target word by pressing the left and right arrow keys of the computer keyboard, respectively. They were required to use their left index finger to press the left arrow key, and the right index finger to press the right arrow key. Participants in the fMRI group responded using the buttons corresponding to thumb and index fingers in an MRI compatible device held in their right hand. Response buttons were not counterbalanced for either group. Intertrial interval was fixed at 1,000 ms in the behavioral study and jittered (with pseudorandom values between 1,500 and 3,000 ms) for testing during fMRI acquisition. At the end of a given phrase, a maximum of 1,500 ms was allowed for participants to respond. RTs were calculated from onset of the last word in the phrase until button press. Only trials with correct responses under 1,000 ms were entered into subsequent analyses. Participants' NAD Effect were calculated as the mean RT difference between unstructured (Random block) and structured (NADs block) trials. A repeated measures ANOVA on participants' RT data with within-participants factors Structure (NADs/Random) and Target (Target/No Target) and Order as a between-participants factor

was initially performed to discard block order effects. A repeated measures ANOVA with factors Structure (NADs/Random) and Target (Target/No Target) was subsequently performed to assess the statistical significance of learning.

## Linear mixed model analysis

In order to assess online SL in the NAD learning task within each experimental group, we used a linear mixed model approach to fit learning slopes reflecting RT gains over trials for the NADs versus Random conditions. The use of mixed models to compare the slope between conditions allows the use of RT data for all trials and participants, which results in a more sensitive measure of the online learning process than a single mean value per participant [15,88].

Analyses were performed using the lme4 [89] and lmerTest [90] packages as implemented in the R statistical language (R Core Team, 2012). Our basic model included RT (*rt*) and trial as continuous variables, *condition* (Random, NADs), and *TNT* (target/no target) as 2-level factors, and participant as a factor with as many levels as participants in each group.

$$rt \sim condition + trial + condition * trial + TNT + (1|participant) \tag{1}$$

$$rt = \text{ß0(intercept)} + \text{ß1}(condition) + \text{ß2}(trial) + \text{ß3}(TNT) + \text{ß4}(condition * trial) \tag{2}$$

As shown in (1), which indicates the specified model, we introduced condition, trial, and their interaction as fixed effects terms. TNT was included as a predictor of no interest. To account for interparticipant variability in basal response speed, we allowed for a different intercept per participant by introducing participant as a random effect. The algebraic expression of the fixed effects part of the model is given in (2). Note that, in this this model, β4 (condition*trial) represents an estimate of the difference in learning slopes between the NADs and Random conditions and can therefore be interpreted as a detrended learning slope estimate for the NADs condition. For the sake of clarity, we have referred to this estimate as βdiff. A statistically significant negative βdiff indicates that online NAD learning effectively took place over and beyond any RT gain that may be attributed to practice effects.

## Temporal difference model

We modeled participants' learning of the dependencies using a TD model [18,19]. Drawing from earlier models of associative learning, such as the RW model [38], the main assumption of TD models is that learning is driven by a measure of the mismatch between predicted and actual outcome [18,19,40,91] (i.e., prediction error (PE)). For instance, when an X element is presented in a NAD block's AXC trial, the PE is computed as:

$$\partial^1 = V(\text{X}^i)_t - V(\text{A}^i)_t \tag{3}$$

where $\partial^1$ is the PE term at element X and trial t, which amounts to the discrepancy between the action value at that state [$V(\text{X}^i)$], and the predictions driven by the previous visited state [$V(\text{A}^i)$].

Computationally, learning through TD is therefore conceptualized (and modeled) as *prediction* learning [40], where the action values/predictions of the previous visited state (following the previous example, of element A) are then updated according to:

$$V(\text{A}^i)_{t+1} = V(\text{A}^i)_t + \alpha \cdot \partial^1 \tag{4}$$

where $\alpha$ is a free parameter that represents the *learning rate* of the participant and determines the weight attributed to new events and the PE they generate [18].

Similarly, when hearing element C, a new PE is generated based on the presence or absence of the target word:

$$\partial^2 = R - V(X^i)_t \tag{5}$$

where R is specified as +1 if it is the target element or −1 if not. Note that the sign choice represents a convenient yet arbitrary means to distinguish target and no target outcomes within the same model. This could have been inverted ($R$ (target) = −1, $R$ (no target) = 1) with no difference in the model's results. Element X is then updated accordingly:

$$V(X^i)_{t+1} = V(X^i)_t + \alpha \cdot \partial^2 \tag{6}$$

One of the advantages of TD models over simpler models of learning, such as the RW (see below), is that they account for the *sequence* of events leading to an outcome, rather than treating each trial as a discrete temporal event. That is, although each trial for the participant (i.e., each 3-word phrase) was equivalently treated as a trial for the TD model, model updates occurred at the presentation of each individual element (see below). TD models are thus sensitive to the precise temporal relationship between the succession of predictions and outcomes that take place in a learning trial [18]. Note that this is particularly valuable in trying to account for the learning of NADs as distinct from adjacent dependencies, making a TD model preferable in such cases. This feature is implemented as a temporal discounting factor; this is an additional free parameter $\gamma$ that represents the devaluation of predictions that are more distant from the outcome [47,92]. Thus, upon "hearing" the final element of a structured (AXC) phrase, the prediction from the initial element A is also updated according to:

$$V(A^i)_{t+1} = V(A^i)_t + \alpha \cdot \partial^2 \cdot \gamma \tag{7}$$

The absolute value of $V(A^i)$ reflects its predictive capacities and the associative strength between element A and a particular response, with higher values indicating stronger predictions. Because of this, have replaced the formal term $V(A)$ by the alternative term $p(A)$ throughout the manuscript, as $p(A)$ may be more intuitively related to *predictions from element A* by the general reader than $V(A)$.

As a behavioral index of participants' predictive capacities, we used RTs, since RTs should be faster when the associations are learnt than when they are not. RTs were first standardized (z scored; zRT) and then normalized between 0 and 1 by the softmax type function:

$$xRT = (1./(1 + \exp(zRTs))) \tag{8}$$

Note that this function will output larger xRT values for lower input zRT values (and, conversely, smaller xRTs values for higher input zRTs), in accord with the idea that better predictive capacities will elicit faster RTs. Importantly, the function also minimizes the effect of extreme RT values.

To fit the free model parameters to each participant's responses, we assumed the following function to minimize the difference between the absolute value of $V(A^i)$ and the transformed RT in a given trial t:

$$\Delta_t = 1 - abs(xRT^i_t - abs(V(A^i)_t)) \tag{9}$$

We then selected the $\alpha$ and $\gamma$ values that produced the maximum LLE, indicating the best possible fit between the model predictions and the participant's transformed RTs. For this, we used Matlab's (Matlab R2017 by Mathworks) *fmincon* function, which implements a Nelder–Mead simplex method [93].

*jupo* — A — $p$(A)

X — $\partial^1 = p(X) - p(A)$
*bade* — $p$(X) — $p(A) = p(A) + \partial^1 * \alpha$

$\partial^2 = R - p(X)$
*runi* — C — $p(X) = p(X) + \partial^2 * \alpha$
$p(A) = p(A) + \partial^2 * \alpha * \gamma$

Trial (t)

**Fig 6. TD model's computations during a NADs block's trial.** The initial A element of the phrase (e.g., *jupo*) carries a prediction value $p$(A). A prediction error $\partial^1$, generated when this prediction is not met by the occurrence of the second word X (*bade*), is used to update the initial prediction $p$(A), scaled by the learning rate $\alpha$. A new prediction is issued from this second word $p$(X), which also generates a prediction error $\partial^2$ on C (*runi*). $\partial^2$ is then used to update both $p$(X) and $p$(A), scaled by the learning rate and (down)scaled by the temporal discounting factor $\gamma$ in the case of the more distant prediction $p$(A). NAD, nonadjacent dependency; TD, temporal difference.

The model was then run for each block (NADs and Random) separately, from which trial-wise prediction values [abs($V$(A)$_t$) and abs($V$(X)$_t$)] for the different phrase elements A and X (resulting in matrices $P$(A) and $P$(X), respectively) were computed.

In summary, adopting the alternative terminology ($p$(A) instead of abs($V$(A))), the TD algorithm for each trial was implemented as depicted in Fig 6.

To illustrate the consistency between participants' RTs and model predictions, both which we assume to be proxies for SL, we plotted the development of $P$(A) computed by the model (inverted as 1-$P$(A)) averaged across participants against the mean RTs of the participants over trials in the NADs block (both z-scored; main text Figs 2 and 3).

To assess the fit of the TD model, we computed for each participant the LLR between the TD model's LLE and the LLE produced by a model predicting at chance. To make fit assessment more intuitive, a model fit index was then calculated as 1 –LLR, where higher model fit index values equate to a better fit. The overall fit of the TD model was assessed at the group level by averaging across participants model fit indexes.

### Rescorla-Wagner model

The RW model was specified as follows: in each trial, the state values $V$(A$^i$)$_t$ and $V$(X$^i$)$_t$ were summed to produce a single prediction $V$(A$^i$X$^i$)$_t$ per trial. No PE was computed on the second element of each phrase (e.g., X in NAD phrases). A PE was computed for each C element as $\partial = R - V(A^i X^i)_t$. This was then used to update both $V$(A$^i$)$_t$ and $V$(X$^i$)$_t$ as $V(A^i)_{t+1} = V(A^i)_t + \alpha \cdot \partial$ and $V(X^i)_{t+1} = V(X^i)_t + \alpha \cdot \partial$, respectively. Note that this is equivalent to the TD model's updates without the $\gamma$ term.

### fMRI acquisition and apparatus

The SL task comprised a single run with 830 volumes. Functional T2$^*$-weighted images were acquired using a Siemens Magnetom TrioTim syngo MR B17 3T scanner and a gradient echo-planar imaging sequence to measure BOLD contrast over the whole brain [repetition time (TR) 2,000 ms, echo time (TE) 25 ms; 35 slices acquired in descending order; slice-thickness: 3.5 mm, $68 \times 68$ matrix in plane resolution = $3.5 \times 3.5$ mm; flip angle = 180°]. We also

acquired a high-resolution 3D T1 structural volume using a magnetization-prepared rapid-acquisition gradient echo (MPRAGE) sequence [TR = 2,500 ms, TE = 3.69 ms, inversion time (TI) = 1,100 ms, flip angle = 90˚, FOV = 256 mm, spatial resolution = 1 mm$^3$/voxel].

## fMRI preprocessing and analysis

Data were preprocessed using Statistical Parameter Mapping software (SPM12, Wellcome Trust Centre for Neuroimaging, University College, London, UK; www.fil.ion.ucl.ac.uk/spm/ ). Functional images were realigned, and the mean of the images was coregistered to the T1. The T1 was then segmented into gray and white matter using the Unified Segmentation algorithm [94], and the resulting forward transformation matrix was used to normalize the functional images to standard Montreal Neurological Institute (MNI) space. Functional volumes were resampled to 2 mm$^3$ voxels and spatially smoothed using an 8-mm FWHM kernel.

Several event-related design matrices were specified for convolutions with the canonical hemodynamic response function. Trial onsets were always defined as the onset of the first word of the phrase. To identify brain regions related to the trial-by-trial development of participants' predictions/PEs, a model with the conditions NADs Target, NADs No Target, Random Target and Random No Target, and all offline test conditions (see S5 Fig) was specified at the first level. This also included, in first place and for each trial for each of the conditions of interest (NADs Target, NADs No Target, Random Target, and Random No Target), a parametric modulator (a vector) corresponding to the RT (z-scored and inverted); and in second, a parametric modulator (a vector) corresponding to the trial-by-trial prediction/PE ($p$(A), also z-scored; S5 Fig). Events were time locked to the onset of the phrase auditorily presented in that trial. In all cases, data were high-pass filtered (to a max. of 1/90 Hz). Serial autocorrelations were also estimated using an autoregressive (AR [1]) model. We additionally included, in all the models described above, the movement parameter estimates for each participant computed during preprocessing to minimize the impact of head movement on the data (S5 Fig).

For each participant, the following contrasts were calculated at the first level (S5 Fig):

- NADs z$P$(A) versus implicit baseline;

- NADs z$P$(A) versus Random z$P$(X1);

- NADs z$P$(A) versus NADs zinvRT;

- NADs z$P$(A) versus Random zinvRT; and

- NADs versus Random.

These were subsequently entered into corresponding one-sample *t* tests *at the second level* to arrive at the reported fMRI results.

We used the Automated Anatomical Labelling Atlas [95] included in the xjView toolbox (https://www.alivelearn.net/xjview/) to identify anatomical and cytoarchitectonic brain areas. Group results are reported for clusters at a $p < 0.001$ FWE-corrected threshold at the cluster level ($p < 0.001$ uncorrected at the voxel level), with a minimum cluster extent of 20 voxels.

## Supporting information

**S1 Fig.** Plot of **(A)** behavioral group and **(B)** fMRI group participants' mean RTs (blue) against the RW model's estimates of the development of predictions over learning (red; inverted as 1-$P$(A) before averaging and z-scoring for display purposes). Vertical bars are the SD. RTs were initially transformed (Materials and methods) and are plotted with the model

prediction estimates in z-score values. *P*(A) = RW model's predictions from the initial word (A) of the dependencies. Data used to generate S1 Fig can be found in S3 Data. fMRI, functional magnetic resonance imaging; RT, reaction time; RW, Rescorla-Wagner.
(DOCX)

**S2 Fig. Brain regions related to changes in the predictive value of the initial word of each phrase in the NADs block (*P*(A)) vs. changes in the predictive value of the initial word of each phrase (*P*(X1)) in the Random block (i.e., *P*(A)-modulated NADs block vs. *P*(X1)-modulated Random block).** Activity in the basal ganglia (bilateral caudate nuclei, putamen, and ventral striatum; see S2 Table) was modulated by the trial-by-trial development of predictions (*P*(A)) as estimated by the TD model. Results are reported for clusters FWE-corrected at $p < 0.001$ at the cluster level (minimum cluster size = 20). Neurological convention is used with MNI coordinates shown at the bottom right of each slice. Data used to generate S2 Fig can be found in http://identifiers.org/neurovault.collection:10421. FWE, family-wise error; NAD, nonadjacent dependency; STG, superior temporal gyrus; TD, temporal difference.
(DOCX)

**S3 Fig. Brain regions related to changes in the predictive value of the initial word of each phrase in the NADs block (*P*(A)) controlling for overt motor response activity in the Random block (i.e., *P*(A)-modulated NADs block vs. RT-modulated Random block).** Subtracting the activity for the Random block modulated by participants' RTs from the *P*(A)-modulated NADs block activity had virtually no effect on basal ganglia activity estimates (see S3 Table). Significant activity centered on the caudate nuclei and the right putamen. Results are reported at a $p < 0.001$ FWE-corrected threshold at the cluster level with 20 voxels of minimum cluster extent. Neurological convention is used with MNI coordinates shown at the bottom right of each slice. Data used to generate S3 Fig can be found in http://identifiers.org/neurovault.collection:10421. NAD, nonadjacent dependency; RT, reaction time.
(DOCX)

**S4 Fig. Brain regions related to changes in the predictive value of the initial word of each phrase in the NADs block (*P*(A)) controlling for overt motor response activity in the NADs block (i.e., *P*(A)-modulated NADs block vs. RT-modulated NADs block).** Significant activations by the contrast between *P*(A)-modulated NADs block and RT-modulated NADs block activity (see also S4 Table) were found in a widespread left-lateralized network of areas, including a large portion of the IFG, parts of the pre- and post-CG, and of the STG in and around the left auditory cortex. Interestingly, bilateral caudate nuclei were also statistically significant along with a small portion of the thalamus. Results are reported for clusters FWE-corrected at $p < 0.001$ at the cluster level (minimum cluster size = 20). Neurological convention is used with MNI coordinates shown at the bottom right of each slice. Data used to generate S4 Fig can be found in http://identifiers.org/neurovault.collection:10421. CG, central gyrus; FWE, family-wise error; IFG, inferior frontal gyrus; NAD, nonadjacent dependency; RT, reaction time; STG, superior temporal gyrus; TTG, transverse temporal gyrus.
(DOCX)

**S5 Fig. fMRI first-level model and contrasts.** Model fit to each participant's fMRI data and vectors used to compute the specified contrasts. fMRI, functional magnetic resonance imaging; NAD, nonadjacent dependency.
(DOCX)

**S1 Table. Whole brain fMRI activity for the NADs *P*(A)-modulated activity vs. implicit baseline contrast.** Group-level fMRI local maxima for the *P*(A)–modulated NADs block

against implicit baseline contrast (see also red-yellow regions in Fig 4, main text). Results are reported for clusters FWE-corrected at $p < 0.001$ at the cluster level (minimum cluster size = 20). MNI coordinates were used. BA, Brodmann area; fMRI, functional magnetic resonance imaging; FWE, family-wise error; NAD, nonadjacent dependency.
(DOCX)

**S2 Table. Whole brain fMRI activity for the *P*(A)-modulated NADs block vs. *P*(X1)-modulated Random block contrast.** Group-level fMRI local maxima for the *P*(A)–modulated NADs block minus *P*(X1)–modulated Random block contrast (see also red-yellow regions in S2 Fig). Results are reported for clusters FWE-corrected at $p < 0.001$ at the cluster level (minimum cluster size = 20). MNI coordinates were used. BA, Brodmann area; fMRI, functional magnetic resonance imaging; FWE, family-wise error; NAD, nonadjacent dependency.
(DOCX)

**S3 Table. Whole brain fMRI activity for the *P*(A)-modulated NADs block vs. RT-modulated Random block contrast.** Group-level fMRI local maxima for the *P*(A)-modulated NADs block minus RT-modulated Random block contrast (see also red-yellow regions in S3 Fig). Results are reported at a $p < 0.001$ FWE-corrected threshold at the cluster level with 20 voxels of minimum cluster extent. MNI coordinates were used. BA, Brodmann area; fMRI, functional magnetic resonance imaging; FWE, family-wise error; NAD, nonadjacent dependency; RT, reaction time.
(DOCX)

**S4 Table. Whole brain fMRI activity for the *P*(A)-modulated NADs block vs. RT-modulated NADs block contrast.** Group-level fMRI local maxima for the *P*(A)-modulated NADs block minus RT-modulated NADs block contrast (see also red-yellow regions in S4 Fig). Results are reported for clusters FWE-corrected at $p < 0.001$ at the cluster level (minimum cluster size = 20). MNI coordinates were used. BA, Brodmann area; fMRI, functional magnetic resonance imaging; FWE, family-wise error; NAD, nonadjacent dependency; RT, reaction time.
(DOCX)

**S5 Table. Whole brain fMRI for the NADs block vs. Random block activity covarying with the NADs effect contrast.** Group-level fMRI local maxima for the NADs block against Random block contrast covarying with the NADs Effect (see Fig 5). Results are reported for clusters FWE-corrected at $p < 0.001$ at the cluster level (minimum cluster size = 20). MNI coordinates were used. BA, Brodmann area; fMRI, functional magnetic resonance imaging; FWE, family-wise error; NAD, nonadjacent dependency; RT, reaction time.
(DOCX)

**S1 Text. Offline recognition test.** Results of the offline recognition test that participants performed after the online incidental NAD learning task. NAD, nonadjacent dependency.
(DOCX)

**S1 Data. Raw data used to create Fig 2.**
(XLSX)

**S2 Data. Raw data used to create Fig 3.**
(XLSX)

**S3 Data. Raw data used to create S1 Fig.**
(XLSX)

**S4 Data. Raw data used to create S1 Text.**
(XLSX)

## Acknowledgments

We thank Sonja A. Kotz and Floris de Lange's Predictive Brain Lab for helpful comments on this work.

## Author Contributions

**Conceptualization:** Joan Orpella, Josep Marco-Pallarés, Ruth de Diego-Balaguer.

**Data curation:** Joan Orpella.

**Formal analysis:** Joan Orpella, Ernest Mas-Herrero, Pablo Ripollés, Josep Marco-Pallarés.

**Funding acquisition:** Ruth de Diego-Balaguer.

**Investigation:** Joan Orpella, Ruth de Diego-Balaguer.

**Methodology:** Joan Orpella, Ernest Mas-Herrero, Josep Marco-Pallarés, Ruth de Diego-Balaguer.

**Project administration:** Ruth de Diego-Balaguer.

**Resources:** Ruth de Diego-Balaguer.

**Software:** Joan Orpella.

**Supervision:** Josep Marco-Pallarés, Ruth de Diego-Balaguer.

**Validation:** Joan Orpella.

**Visualization:** Joan Orpella.

**Writing – original draft:** Joan Orpella, Ernest Mas-Herrero, Pablo Ripollés, Josep Marco-Pallarés, Ruth de Diego-Balaguer.

**Writing – review & editing:** Joan Orpella, Ernest Mas-Herrero, Pablo Ripollés, Josep Marco-Pallarés, Ruth de Diego-Balaguer.

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
