## [Editor Report · Decision Letter 0]

25 Jan 2021

Dear Dr de Diego-Balaguer, 

Thank you for submitting your manuscript entitled "Statistical learning as reinforcement learning phenomena" for consideration as a Research Article by PLOS Biology.

Your manuscript has now been evaluated by the PLOS Biology editorial staff, as well as by an academic editor with relevant expertise, and I am writing to let you know that we would like to send your submission out for external peer review.

Please re-submit your manuscript within two working days, i.e. by Jan 27 2021 11:59PM.

Kind regards,

Gabriel Gasque, Ph.D.,

Senior Editor

PLOS Biology

---

## [Decision Letter · Decision Letter 1]

4 Mar 2021

Dear Dr de Diego-Balaguer,

Thank you very much for submitting your manuscript "Statistical learning as reinforcement learning phenomena" for consideration as a Research Article at PLOS Biology. Your manuscript has been evaluated by the PLOS Biology editors, by an Academic Editor with relevant expertise, and by three independent reviewers.

In light of the reviews (below), we will not be able to accept the current version of the manuscript, but we would welcome re-submission of a much-revised version that takes into account the reviewers' comments. We cannot make any decision about publication until we have seen the revised manuscript and your response to the reviewers' comments. Your revised manuscript is also likely to be sent for further evaluation by the reviewers.

We expect to receive your revised manuscript within 3 months. 

**IMPORTANT - SUBMITTING YOUR REVISION**

Your revisions should address the specific points made by each reviewer. As you will see, all the reviewers raised a number of points of clarification that will have important impacts on how the data can be interpreted. Reviewers 1 and 2 also make important conceptual comments in their points 1.1 and 2, respectively. The points acknowledge that the results might constitute evidence for the existence of prediction effects but ask if the prediction effects observed might actually be consistent with several models of learning and might even be the consequence of learning in a statistical learning mechanism. Reviewer 2, in their point 1, also argues that because the task is deterministic rather than probabilistic, the prediction errors that are generated in the task are always in one direction and so it is not clear that both RT increases and decreases would be predicted during learning. It will be particularly important to address these points if you decide to revise the manuscript.

Please submit the following files along with your revised manuscript:

*Re-submission Checklist*

*Published Peer Review*

*PLOS Data Policy*

*Blot and Gel Data Policy*

Sincerely,

Gabriel Gasque, Ph.D.,

Senior Editor,

ggasque@plos.org,

PLOS Biology

REVIEWS:

Reviewer #1: This work applies a reinforcement learning model (a Temporal Difference model, TD) to account for behavioral and neural measures of Statistical Learning (SL) of non-adjacent dependencies. Data from two studies are reported. First is a behavioral experiment showing that the TD model predicts change in RTs over repetitions in a structured block of the SL task. The second is an fMRI study, including in-scanner behavioral data replicating the first experiment, plus showing that the model's parameter reflecting the extent of prediction associated with the first element within patterns is correlated with trial-level activation in the striatum. Together, the authors interpret these two pieces of data as support for a strong mechanistic role of prediction in SL processes. 

My overall impression of the manuscript is positive. As someone who works on SL but has limited knowledge of reinforcement learning and its models, I found the manuscript to be novel and thought-provoking. The combination of computational modeling, behavioral data from online SL measures, and neuroimaging should also be applauded. At the same time, there are issues in this version of the manuscript that should be addressed. Below I start by reviewing conceptual issues, followed by methodological ones. 

1. Conceptual issues: The role of prediction in SL. 

1.1 As mentioned above, two pieces of data support the link between prediction and SL: The TD model's ability to predict change in RT behaviorally and the correlation between the model's parameter reflecting prediction and activation in the striatum. Both points convincingly demonstrate involvement of prediction in the SL task, which is already an important contribution. What these findings do not show, however, is that prediction is the mechanism responsible for SL (which is what the authors argue throughout). Rather, both types of evidence are consistent with SL models where prediction is the consequence, not the driver. Specifically, the fact that a TD model performs better than chance (or better than a simple Rescorla-Wagner model that is not set to learn non-adjacent patterns) is consistent with theories where prediction is the consequence (more SL -> more predictions -> reduced RTs). Similarly, the fMRI evidence is also consistent with prior work showing striatal involvement in SL (which again can be a consequence of learning). What is missing from the paper is a 'smoking gun' type of evidence, which ties SL performance directly to extent of prediction. For example, an analysis showing that a substantial part of the inter-individual variance in the SL task can be traced back to model's parameters/striatal involvement (rather than that there is striatal involvement, but that most of the variance is carried by activation in, say, the hippocampus- see next point). If such additional evidence is not available, the authors need to be more careful in both the framing and discussion/conclusions.

1.2 A more detailed account of how the current theory/model differs from other models of SL is needed. Specifically, the authors should consider recent models that tie SL to the hippocampus (Schapiro et al., 2017 is a good example for a neurobiologically plausible computational work), as well as to network-level representation of uncertainty (Hasson, 2017). Is the current model meant to replace these models or complement them? What does the current model provide that these models do not? What type of evidence is needed to adjudicate between the current model and these previous attempts? In general, and in the same line of thought as the point above, any direct evidence of the current model's added value compared to existing models would further increase the impact of this paper. 

1.3 Relatedly, I'd appreciate a discussion of the relation between the current work and models that take part in SL phenomena within other domains. For example, in the context of reading, connectionist models that use prediction errors to learn the input structure are widely used (starting from Seidenberg & McClelland, 1989), and these models are now increasingly viewed as participating in SL of the written input (see Sawi & Rueckl, 2019; Seidenberg, 2011). Are there ways in which the current model is conceptually different from such connectionist models, which are used to learn statistical regularities outside 'typical' SL tasks? 

2. Methodological issues: Clarification of computational and statistical models.

2.1 The description of the methods is generally clear, except for the description of how RTs in the task were used to derive the model's parameters. On p. 18, it is stated that "[t]o obtain the free parameters for each participant … [w]e selected the values that produced the minimum Log-Likelihood Estimate (LLE), indicating the best possible fit between the model predictions and the participant's transformed RTs." This is unclear. What was the function that linked the model's parameters to the RT data? 

2.2 The statistical approach used in the fMRI analysis is underspecified (and potentially flawed). It seems that the authors used a 'simple' GLM to examine the relation between trial-level estimated parameters (from the model) and the trial-level activation (after controlling for additional factors). But, trials are nested within subjects (i.e., multiple trials with different values within subjects), and thus a simple GLM may not be appropriate. If this is indeed the case, please correct and use an a statistical method that deals with the nesting of trials within subjects (e.g., mixed effect models; see Chen et al. 2013). If my reading of the text is incorrect, please clarify.

References

Chen, G., Saad, Z. S., Britton, J. C., Pine, D. S., & Cox, R. W. (2013). Linear mixed-effects modeling approach to FMRI group analysis. Neuroimage, 73, 176-190.

Hasson, U. (2017). The neurobiology of uncertainty: implications for statistical learning. Philosophical Transactions of the Royal Society B: Biological Sciences, 372(1711), 20160048.

Sawi, O. M., & Rueckl, J. (2019). Reading and the neurocognitive bases of statistical learning. Scientific Studies of Reading, 23(1), 8-23.

Schapiro, A. C., Turk-Browne, N. B., Botvinick, M. M., & Norman, K. A. (2017). Complementary learning systems within the hippocampus: a neural network modelling approach to reconciling episodic memory with statistical learning. Philosophical Transactions of the Royal Society B: Biological Sciences, 372(1711), 20160049.

Seidenberg, M. S. (2011). Reading in different writing systems: One architecture, multiple solutions.

Seidenberg, M. S., & McClelland, J. L. (1989). A distributed, developmental model of word recognition and naming. Psychological review, 96(4), 523.

Reviewer #2: This paper asks about the dynamic of statistical learning and its neural bases presenting first a behavioral experiment and then an fMRI one of the same task. Participants were asked to respond as quickly as possible to target pseudo-words. The predictability of these targets was manipulated within subjects between two task versions. While the three pseudo-words of each trial phrase were random in a control version, in the learning version the first pseudo-word was perfectly predictive of the last pseudo-word of each phrase. In that predictive version, the participants' response times to the final word elements decreased as they learned to anticipate the final element from the predictive first element. A reinforcement learning algorithm was fit to participants' data, with the resulting fits being used to analyze behavioral data and extract learning parameters, which were then related to BOLD signal. Primary analyses probed the relations between trials' BOLD signal and reinforcement learning algorithm fits. The authors conclude that a Temporal Difference model provides a good algorithmic description of learning during the predictive task. The authors then further present an algorithmic interpretation of neural correlates of learning including striatal activity.

While providing a rich and interesting read, this paper suffers from two major weaknesses that should be addressed before it can be considered fit for publication:

1. The authors state in the abstract and introduction that their focus is on revealing the neural bases of statistical learning as it unfolds but their paper suffers throughout from a rather fuzzy use of the terms "statistical learning," "rule learning", and "reinforcement learning." Mechanistically, the literature points to these various forms of learning as having distinct characteristics. Yet, it remains unclear to this reader whether this paper focuses on statistical learning or another form of learning. Indeed, the actual predictive task used by the authors is best described (as the authors do in their Methods) as "rule learning.", and not statistical learning. Participants simply have to learn a one-to-one, deterministic correspondence between a cue and a response. The learning challenge is made less trivial by the insertion of a long-distance dependency - yet it seems best characterized as an instance of associative learning, in this case entirely deterministic and quite unlike what is labeled statistical learning in the literature referenced by the authors. To the extent that a purely deterministic system's contingencies do not have statistical variation, the authors' claims about many putative processes, such as prediction error, appear to be at odds with the actual task completed by participants. While the authors claim that the close correspondence between the TD fits and the RT are evidence for particular mechanisms, it seems likely that the TD model is simply very flexible and is able to take the shape of the data. For example, it is not clear why change in RT would be nonmonotonic, as shown in Figures 1 and 2 (i.e., sometimes increasing with learning). The TD model would predict that RT should increase when expectations are violated. But, in a deterministic system with fairly simple rules, the participants' expectations were never violated by the stimuli. As the core novelty of this paper is based on applying reinforcement learning algorithm to a task situation of statistical learning, a task that consists of learning deterministic associations would be entirely at odd with the paper stated goal. How the main learning task consists into a form of statistical learning should be clarified.

2. The authors' framing of their field, and their results in that field, repeatedly refer to the potential for bidirectional causality between changes in predictions and changes in statistical learning. This point would benefit from being carefully argued for. The authors never explain why statistical learning could not be instantiated by learning to predict; that is, the change in prediction *is* the learning or in other words, diagnostic of the behavior of a system demonstrating the presence of learning. The authors describe their trial-by-trial estimates of predictions, and clearly they have trial-by-trial BOLD measurements. Given the focus of the authors on the causal directionality of prediction vs. learning, it seems analyses examining the temporal order of the development of BOLD measurements in relation to the development of the prediction estimates should be in order at a finer-grained timescale than aggregating across their entire blocks as they do. Using a more precise timescale of analysis may help the authors in other ways as well. For example, the results in line 140 are surprisingly weak, which may have been because the authors averaged across the entire behavioral blocks. If the intention was to test for learning, rather than performing analysis by block, the authors could show that RTs decrease at a steeper rate and to a greater extent in the predictive than random block. This particular problem may also be addressed in part by using mixed-effects models with by-participant random effects rather than averaging within participants and then using ANOVA.

Apart from these two main comments, we have a number of other questions about the work:

3. Another source of worry from this reader concerns the definitions of clusters using a whole-brain searchlight using the test statistics from p(A) as a predictor within a mixed-effects model. The minimum cluster size and related details are mentioned line 491, but beyond this brief text, it's not really clear how the earlier explanations (e.g., discussing controlling for motor response) relate to the actual cluster definitions. In particular, when lines 167-168 say "we correlated this proxy for prediction learning with participants' trial-wise BOLD signal measures for the Rule block" it is difficult for the reader to know what is actually happening in terms of model fitting/assessment. While the Result section is indeed not the place to thoroughly explain such a procedure, the authors could ensure that an interested reader can determine exactly how this "correlation" was implemented.

4. The authors mention that TD models may be particularly able to capture the computational features of learning associated subcortical activity (lines 303-304 - "a possible explanation for this concerns the emphasis that particular models of learning place on specific computations, with TD models being most sensitive to subcortical (rather than neo-cortical) activity."). In fact, the authors have the ability to test this, don't they? It would seem appropriate to back up their statement with a model comparison between TD and other simpler models, such as RW in both subcortical and neocortical areas to properly document their point. If this particular contrast (i.e., between TD and other models' neural analogues) is not what the authors were referring to in line 304, then they should clarify.

5. A couple points regarding model fitting:

o Why were z-scored RTs exponentiated prior to normalizing between 0 and 1 (line 419)? Does this pertain to the specific distributional assumptions of the TD model? RT distributions are typically quite skewed, and the authors note that there are "fluctuations due to decision processes, motor response preparation and execution, random waxing and waning of attention, and system noise". This is true, and these are reasons that rare large RTs have disproportionate effect on, say, mean RT values. In fact, RTs are often log-transformed in order to better approximate normality (although this procedure has its own problems), but exponentiating the distribution of RTs would have the opposite effect, and only *increase the influence* of the rare and large RT data points. 

o On line 420, did the authors meat that their parameter optimization found the maximum log-likelihood estimate? The minimum log-likelihood is probably not defined for their function (if their optimization problem is typically concave), nor would it be theoretically interesting. A common approach to maximum-likelihood estimation is to minimize the negative log-likelihood; it's possible the authors reversed this in their explanation.

6. Figure captions should clearly indicate the measure of variability. For example, in Figure 2, the caption describes "mean reaction times and error bars" and later in the caption it says that "vertical bars are the SD." However, I'm guessing that the initial statement about error bars does not refer to SD, but refers to either standard errors or confidence intervals.

7. "Welcome Trust" (line 458) should be "Wellcome Trust"

Reviewer #3: Review for "Statistical learning as reinforcement learning phenomena" by Orpella and co. The authors aimed at investigating the computational and neural correlates of statistical learning in the context of artificial language acquisition. The run two experiments (one behavioural, the other fMRI) and found that statistical learning (as indexed by reaction times) is well explained by a temporal difference prediction error model (compared to a random model and a Rescorla-Wagner model) and that TD prediction errors are represented in the network encompassing the striatum and the ventral prefrontal cortex. The paper is very clear and the results seem robust. The topic is of interest for broad readership and I praise the authors for assessing the too-much-neglected role of reinforcement learning processes in language learning. I have several suggestions that I think could improve the paper. In each categories, the issues are ranked by order of importance. 

-- behavioral task and results 

1/ I strongly encourage the authors to include a figure with the task / stimuli. For people not working in language (as me) it is very hard to figure out what the subjects are actually seeing / listening to / doing. This should be the first figure of the paper and I also recommend to clearly specify to which time steps of the task the functional activations correspond to. 

2/ I strongly encourage the authors to report not only the mean ± sem (barplots), but also the individual points. It is becoming standard in the field and for good reasons. 

-- fMRI

1/ My main issue concerning the fMRI results is the lack of clarity concerning the exact structure of the GLMs and the contrast used. I strongly encourage the authors to include a clear presentation of the GLMs and the contrast used, similar to this one: 

https://www.jneurosci.org/content/jneuro/suppl/2009/10/28/29.43.13465.DC1/Supplemental_Material.pdf

2/ Related to the previous issue, the manuscript lacks clarity concerning the preprocessing (z scoring or not?) of the parametric modulators and which parameters (individual or group level) have been used to generate the regressors. Please refer to Lebreton et al. Nature Human Behaviour (2019) for a full treatment of this issue in the context of model-based fMRI. 

3/ There are many interesting fMRI results (for example P(A) - Rule or P(A) - Norule) that are currently only presented in the SI. I believe they deserve a place in the supplementary materials. Also (see my previous point) I am not sure whether the authors report the basic reaction time contrast. Were the correlates of reaction times different across conditions? 

4/ I also encourage the authors to check for the negative correlations of the prediction errors (there is an hypothesis about different neural systems processing PEs in opposite directions), for example in the insula or dorsomedial prefrontal cortex. 

5/ in addition to the activation tables, the authors should consider uploing the SPMs into neurovault.org

6/ Concerning fMRI reporting, the authors should explicit the significance threshold used for the fMRI results in the Results sections ('lines 164 onward'). In the current version it is only specified in the methods section and in the legend. 

-- computational modeling 

1/ The notation used by the authors is really not-standard for the TD framework. I actually had trouble convincing myself that the model was roughly equivalent to what I know as the TD-learning from Sutton and Barto's book. The authors should consider rewriting the formalism in a more standard way. 

2/ The authors should also explicit the equations of the RW model. I assumed it's equivalent to the TD, but with Gamma=0? A non-expert will need this information. 

-- discussion

1/ The authors should consider citing / discussing Giavazzi et al Cortex (2018), which is, I believe, consistent with their caudate results. 

-- typos

1/ Line 248 "oft-reported"

---

## [Editor Report · Decision Letter 2]

15 Jul 2021

Dear Dr de Diego-Balaguer,

Thank you for submitting your revised Research Article entitled "Statistical learning as reinforcement learning phenomena" for publication in PLOS Biology. I have now discussed your response to reviewers and your new version with the Academic Editor. I am pleased to let you know that we will probably accept this manuscript for publication, pending you addressing the data and other policy-related requests listed below my signature.

In addition, together with the Academic Editor, we think it would be extremely useful for future readers to make the peer-review history of your manuscript accessible, so they can read and assess your thoughtful answers to the reviewers. You will be offered this option later on during the production process, and we ask that you take it.

As you address the items above, please take this last chance to review your reference list to ensure that it is complete and correct. If you have cited papers that have been retracted, please include the rationale for doing so in the manuscript text, or remove these references and replace them with relevant current references. Any changes to the reference list should be mentioned in the cover letter that accompanies your revised manuscript.

We expect to receive your revised manuscript within two weeks. 

*Published Peer Review History*

*Early Version*

Sincerely,

Gabriel Gasque, Ph.D.,

Senior Editor,

ggasque@plos.org,

PLOS Biology

TITLE:

We would like to suggest a more informative/decompressed title: “Language statistical learning responds to reinforcement learning principles rooted in the striatum.”

We would be happy to work with you on an alternative, if you disagree or feel the our proposed title misrepresents your findings. 

ETHICS STATEMENT:

Please include within your manuscript the ID number of the protocol approved by the Universitat de Barcelona and the ERC ethics scientific office.

DATA POLICY:

Note that we do not require all raw data. Rather, we ask for all individual quantitative observations that underlie the data summarized in the figures and results of your paper. For an example see here: http://www.plosbiology.org/article/info%3Adoi%2F10.1371%2Fjournal.pbio.1001908#s5

These data can be made available in one of the following forms:

Regardless of the method selected, please ensure that you provide the individual numerical values that underlie the summary data displayed in the following figure panels: Figures 2AB, 3AB, 4, 5, S1AB, S2, S3, S4, and S1 Text.

Please also ensure that each figure legend in your manuscript includes information on where the underlying data can be found and that your supplemental data file/s has/have a legend.

DATA NOT SHOWN?

---

## [Editor Report · Decision Letter 3]

2 Aug 2021

Dear Dr de Diego-Balaguer,

On behalf of my colleagues and the Academic Editor, Matthew Rushworth, I am pleased to say that we can in principle offer to publish your Research Article "Language statistical learning responds to reinforcement learning principles rooted in the striatum" in PLOS Biology, provided you address any remaining formatting and reporting issues. These will be detailed in an email that will follow this letter and that you will usually receive within 2-3 business days, during which time no action is required from you. Please note that we will not be able to formally accept your manuscript and schedule it for publication until you have made the required changes.

**IMPORTANT:

1) As mentioned previously, we think it would be extremely useful for future readers to make the peer-review history of your manuscript accessible, so they can read and assess your thoughtful answers to the reviewers. You will be offered this option later on during the production process, and we ask that you take it.

2) I have modified your Data Availability Statement in the submission system to say, "All relevant data can be found in the paper's Supporting Information files and in https://neurovault.org/collections/10421/". Let me know if you disagree or have questions or concerns.

PRESS

Sincerely, 

Gabriel Gasque, Ph.D. 

Senior Editor 

PLOS Biology

ggasque@plos.org